# Analysis of Risk Factors for In-Hospital Death Due to COVID-19 in Patients Hospitalised at the Temporary Hospital Located at the National Stadium in Warsaw: A Retrospective Analysis

**DOI:** 10.3390/ijerph19073932

**Published:** 2022-03-25

**Authors:** Sławomir Butkiewicz, Artur Zaczyński, Michał Hampel, Igor Pańkowski, Robert Gałązkowski, Patryk Rzońca

**Affiliations:** 1Emergency Department, Central Clinical Hospital of the Ministry of the Interior and Administration in Warsaw, 137 Wołoska St., 02-004 Warsaw, Poland; slawomir.butkiewicz@cskmswia.gov.pl (S.B.); igor.pankowski@cskmswia.gov.pl (I.P.); 2Clinical Department of Neurosurgery, Central Clinical Hospital of the Ministry of the Interior and Administration in Warsaw, 137 Wołoska St., 02-004 Warsaw, Poland; artur.zaczynski@cskmswia.gov.pl; 3Department of Gastroenterological and Transplant Surgery, Central Clinical Hospital of the Ministry of the Interior and Administration in Warsaw, 137 Wołoska St., 02-004 Warsaw, Poland; michal.hampel@cskmswia.gov.pl; 4Department of Emergency Medical Services, Faculty of Health Sciences, Medical University of Warsaw, 14/16 Litewska St., 00-575 Warsaw, Poland; robert.galazkowski@wum.edu.pl; 5Department of Human Anatomy, Faculty of Health Sciences, Medical University of Warsaw, 5 Chałubińskiego St., 02-004 Warsaw, Poland

**Keywords:** SARS-CoV-2, temporary hospital, COVID-19, mortality, risk factors

## Abstract

The outbreak of the SARS-CoV-2 (severe acute respiratory syndrome coronavirus 2) pandemic has affected all aspects of social life and brought massive changes to the healthcare sector. The aim of this study was to identify the factors affecting the mortality of COVID-19 patients at a temporary hospital in Warsaw (Poland). The present study was conducted based on a retrospective analysis of the medical records of patients hospitalised at the temporary hospital located at the National Stadium in Warsaw between 1 March 2020 and 30 April 2021. The study included all cases of patients who were brought directly or transferred to the National Hospital from other hospitals for further treatment. With regard to comorbidities, the analysis found that five comorbidities—namely, diabetes (OR = 1.750, 95% CI: 1.009–2.444, *p* < 0.05), stroke history (OR = 2.408, 95% CI: 1.208–4.801, *p* < 0.05), renal failure (OR = 2.141, 95% CI: 1.052–4.356, *p* < 0.05), chronic obstructive pulmonary disease (OR = 2.044, 95% CI: 1.133–3.690, *p* < 0.05) and heart failure (OR = 1.930, 95% CI: 1.154–3.227, *p* < 0.05)—had a significant impact on the survival of COVID-19 patients. The analysis identified 14 factors that had a significant impact on the prognosis and mortality of the COVID-19 patients studied.

## 1. Introduction

The emergence of a new pathogen causing acute respiratory disease of an unknown aetiology and first identified at the end of December 2019 in China led to a global pandemic, which was announced as such by the World Health Organization on 11 March 2020. The pathogen was identified as a highly contagious coronavirus and named severe acute respiratory syndrome coronavirus 2 (SARS-CoV-2), which, in humans, causes coronavirus disease 2019 (COVID-19) [1,2,3]. More than 414 million infections and over 5.8 million deaths from COVID-19 have been reported worldwide since the start of the pandemic [4]. The outbreak of the SARS-CoV-2 pandemic has affected all aspects of social life and brought massive changes to the healthcare sector, including those concerning the services provided and the protection of other patients and healthcare professionals against infections [5].

SARS-CoV-2 infections are not always symptomatic. The literature reports many cases of individuals who tested positive but had no clinical symptoms of COVID-19. According to the estimates of researchers, the proportion of asymptomatic individuals who test positive ranges widely, from 17.9% to 57% [6,7,8]. Furthermore, clinical manifestations of COVID-19 include a diverse range of symptoms. Most individuals with a COVID-19 infection (approximately 81%) have a mild disease with either no pneumonia or mild pneumonia. In 14% of cases, COVID-19 presents with severe symptoms such as dyspnoea, hypoxemia and lung infiltrates greater than 50% on radiological imaging. Approximately 5% of patients develop a critical illness that is associated with acute respiratory failure, shock and multiple organ dysfunction or failure [9].

Several risk factors that are predisposed to severe COVID-19 disease have been identified in the literature. These include an older age [10,11,12,13,14], male sex [15,16] and the presence of comorbidities, including cardiovascular disease, diabetes, chronic obstructive pulmonary disease (COPD) and cancer [17,18,19,20,21,22,23]. The most commonly reported symptoms of COVID-19 are a cough, fever, myalgia and a headache. Other well-documented symptoms include diarrhoea, a sore throat and loss/change in the sense of smell or taste. Unfortunately, none of the symptoms listed above are sufficiently specific to make a reliable diagnosis of a SARS-CoV-2 infection [24].

Severe illness from the new coronavirus is associated with complications. The most commonly reported ones include acute respiratory distress syndrome (ARDS) and cardiovascular, thromboembolic, neurological and inflammatory complications as well as secondary infections. Therefore, it is clear that COVID-19 is not solely a respiratory disease that may lead to acute respiratory distress. A range of manifestations have been reported in COVID-19 patients, including cardiac arrhythmias, myocardial damage, heart failure, venous thromboembolism, extensive deep vein thrombosis, pulmonary embolism, encephalopathy, motor and sensory deficits and seizures [25,26,27,28,29]. The above observations further highlight the scale of the problem and the dangers faced by those working in the healthcare sector.

Another important issue related to the coronavirus pandemic, which resulted in a rapid surge of new patients, is the organisation of healthcare. One of the strategies for preparing the healthcare system for a large number of patients is to set up temporary hospitals that are intended to take patients if traditional hospitals have no capacity to treat them. This enables fully equipped traditional hospitals to focus on the most severely affected patients and those who are quickly deteriorating. The literature on the subject stresses that the set-up and use of temporary hospitals is the best way to prepare the healthcare system for a large and rapid increase in COVID-19 patients [30,31,32,33,34,35]. A total of 36 temporary hospitals were set up in Poland from the start of the SARS-CoV-2 pandemic. The first temporary hospital of Poland, called the National Hospital, was set up at the National Stadium in Warsaw. At the peak of the third wave of the pandemic, the hospital had 550 beds and the potential to increase its bed capacity to 1200 beds in a short time [36].

In light of the above, the aim of this study was to identify the factors affecting the mortality of COVID-19 patients at this temporary hospital in Warsaw (Poland).

## 2. Materials and Methods

### 2.1. Study Design

The present study was conducted based on a retrospective analysis of the medical records of patients hospitalised at the temporary hospital located at the National Stadium in Warsaw between 1 March 2020 and 30 April 2021. The study included all cases of patients who were brought directly or transferred to the National Hospital from other hospitals for further treatment. The only exclusion criterion was a gap in the medical records regarding the data required for the analysis. A total of 1749 cases meeting the established criteria were included in the final analysis.

The documentation was analysed in order to obtain the following information: age and sex of the patients, date of admission and discharge, clinical parameters of the patients, medical procedures performed, laboratory test results, clinical symptoms and comorbidities, the MEWS score, ICU transfer information and death information.

### 2.2. Statistical Analysis

The data obtained were statistically analysed using STATISTICA software, version 13.2 (Tibco Software Inc., Palo Alto, CA, USA). The qualitative data were reported using numbers (n) and percentages (%) whereas the quantitative data were described using medians (Me) and interquartile ranges (IQR). The normality of the distribution of the quantitative variables was tested using the Kolmogorov–Smirnov test and the Lilliefors test. The chi^2^ test was used to analyse the statistically significant differences between the qualitative variables whereas the Mann–Whitney U-test was used to analyse the differences between the two independent groups. The impact of particular variables on patient survival was assessed with multivariate logistic regression using a stepwise selection procedure. A significance level of *p* < 0.05 was used in the study.

### 2.3. Ethics

The study was conducted in accordance with the principles of the Declaration of Helsinki. The reports with data were anonymous and did not permit the identification of individual patients at any stage of the study. The study protocol was submitted to the Bioethics Committee at the Medical University of Warsaw, which confirmed that the study did not require consent due to its retrospective nature (AKBE/13/2022).

## 3. Results

In the period analysed, there were 1749 patients at the temporary hospital located at the National Stadium in Warsaw. Most patients were men (59.1%) and their median age was 64 years (IQR 50.0–73.0 years). The median BMI of the patients was 29.0 (IQR 25.9–31.7). Most patients lived in urban areas (85.7%). The most commonly reported symptoms were a cough (58.0%), fever (50.9%) and dyspnoea (50.3%) whereas the most common comorbidities were hypertension (39.4%) and diabetes (15.1%). Patients were most commonly admitted to the temporary hospital between 7:00 AM and 6:59 PM (55.3%). A total of 53 patients (3.0%) had to be transferred to an ICU and 156 patients (8.9%) died. The details are presented in Table 1.

Our analysis showed that, compared with patients who survived hospitalisation and were discharged, patients who died from COVID-19 tended to be older (72.4 years vs. 63.0 years), presented with fever (43.0% vs. 51.7%) and muscle pain (15.4% vs. 24.1%) less often and more often presented with such comorbidities as hypertension (46.8% vs. 38.7%), diabetes (23.7% vs. 14.3%), heart failure (14.7% vs. 6.3%), COPD (10.9% vs. 4.5%), cancer (7.1% vs. 3.5%), renal failure (9.6% vs. 2.9%) and a stroke history (7.7% vs. 3.0%) (*p* < 0.05). Moreover, the difference in the mortality rate between patients who were transferred to an ICU and those hospitalised at the temporary hospital (27.6% vs. 0.6%) was statistically significant (*p* < 0.05). A detailed analysis is shown in Table 1.

The median duration of hospitalisation was 10 days (IQR 7.0–15.0). The median volume of lung parenchyma affected by inflammatory changes indicating COVID-19 severity in the patients studied was 20% (IQR 9.0–35.0) and the median MEWS score was 3 (IQR 2.0–5.0%). The median body temperature of the patients studied was 36.8 °C (IQR 36.4–38.5 °C), the median heart rate was 80 beats/min (IQR 70.0–90.0), the median mean arterial pressure was 96.3 mmHg (IQR 87.0–104.8), the median respiratory rate per minute was 18 (IQR 17.0–21.0) and the median oxygen saturation level was 96% (IQR 94.0–98.0) (Table 2).

Table 2 shows a statistical analysis between COVID-19 patients who survived hospitalisation and those who died during hospitalisation. The analysis found that in patients who did not survive, the median duration of hospitalisation was significantly longer (14.5 days vs. 10.0 days), the volume of lung parenchyma affected by inflammatory changes was greater (30.0% vs. 20.0%) and the median MEWS score was significantly higher (5.0 vs. 3.0) compared with the surviving patients. Moreover, patients who died more often required Optiflow treatment (52.6% vs. 31.5%), endotracheal intubation (28.1% vs. 11.3%) and mechanical ventilation (26.3% vs. 7.9%). An analysis of the laboratory blood test results found that patients who died from COVID-19 had a higher median WBC count (7.9 thousand/µL vs. 6.1 thousand/µL) and neutrophil count (7.0 thousand/µL vs. 4.2 thousand/µL) and a lower median lymphocyte count (0.6 thousand/µL vs. 1.1 thousand/µL), monocyte percentage (4.9% vs. 7.7%) and eosinophil percentage (0.1% vs. 0.1%). The associations identified were statistically significant (*p* < 0.05).

A logistic regression analysis was then carried out to identify the risk factors for in-hospital death due to COVID-19. The analysis showed that admission to an ICU (OR = 60.238, 95% CI: 29.495–123.024, *p* < 0.001), the MEWS score (OR = 1.499, 95% CI: 1.335–1.684, *p* < 0.001) and the need for mechanical ventilation (OR = 2.431, 95% CI: 1.094–5.402, *p* < 0.05) were significant factors affecting the survival of COVID-19 patients. As for comorbidities, the analysis showed that diabetes (OR = 1.750, 95% CI: 1.009–2.444, *p* < 0.05), renal failure (OR = 2.141, 95% CI: 1.052–4.356, *p* < 0.05), COPD (OR = 2.044, 95% CI: 1.133–3.690, *p* < 0.05), a stroke history (OR = 2.408, 95% CI: 1.208–4.801, *p* < 0.05) and heart failure (OR = 1.930, 95% CI: 1.154–3.227, *p* < 0.05) had a significant impact on the survival of COVID-19 patients. With regard to the clinical symptoms, significant factors affecting the mortality of COVID-19 patients were fever (OR = 1.703, 95% CI: 1.504–1.979, *p* < 0.05) and shallow breathing (OR = 1.779, 95% CI: 1.094–2.895, *p* < 0.05). With regard to the clinical parameters, a significant association was found between in-hospital mortality due to COVID-19 and parameters such as heart rate (OR = 1.058, 95% CI: 1.022–0.096, *p* < 0.05) and respiratory rate (OR = 1.129, 95% CI: 1.008–1.263, *p* < 0.05). Other significant factors identified were the haematocrit level (OR = 0.742, 95% CI: 0.630–0.873, *p* < 0.001) and lymphocyte count (OR = 0.601, 95% CI: 0.501–0.721, *p* < 0.001) (Table 3).

## 4. Discussion

The global SARS-CoV-2 pandemic has affected all aspects of life, resulting in fear and social isolation, and has led to numerous changes in healthcare systems. As COVID-19 poses a global threat to public health, research investigating the characteristics of patients infected with the new pathogen as well as the risk factors for infection, severe disease and death from COVID-19 occupies an important place among studies concerning the SARS-CoV-2 pandemic. Such research provides necessary information used to develop strategies for the appropriate treatment of patients [37,38,39,40,41]. Therefore, the aim of this study was to identify the factors affecting the mortality of COVID-19 patients at a temporary hospital located at the National Stadium in Warsaw (Poland).

A total of 1749 cases of patients hospitalised with a SARS-CoV-2 infection at the temporary hospital in Warsaw were analysed. A detailed analysis of the clinical characteristics of the patients identified several factors associated with the COVID-19 prognosis. We found an association between older age and increased mortality from COVID-19, which was consistent with the findings from studies by, for example, Yang et al. (2020), Zhou et al. (2020), Grasselli et al. (2020), Baena-Díez et al. (2020), Vila-Corcoles et al. (2021), Wang and Wang (2021) and Alharthy et al. (2021) [18,40,41,42,43,44,45].

The analysis of our findings indicated that transfer to an ICU, the need for mechanical ventilation and a higher MEWS score were associated with an increased likelihood of death in the patients studied. These findings were consistent with those from studies by Sadeghi et al. and Ughi et al., which found that ICU admission was associated with a higher risk of mortality of COVID-19 patients [46,47]. In their study, Tu et al. demonstrated that the mortality of COVID-19 patients who required mechanical ventilation was high and that a delayed mechanical ventilation initiation was an independent predictor of the 28-day mortality of COVID-19 patients [48]. The findings from a study by Aygun and Eraybar showed that a group of COVID-19 patients who died had a significantly higher MEWS compared with a surviving group [49].

Research on COVID-19 places considerable focus on identifying and analysing the risk factors for death from COVID-19 related to comorbidities [41,42,50,51,52,53] and symptoms [41,52,54], which was also the subject of the present study. Fang et al. found that chronic kidney disease significantly contributed to the death of COVID-19 patients [50]. Similarly, Zolotov et al. found that mortality among COVID-19 patients with an abnormal kidney function was three times higher compared with patients with a normal kidney function [55]. A study by Li et al. (2021) demonstrated that diabetes, immunosuppression and cancer were most strongly associated with severe illness from COVID-19. Moreover, the authors found an association between hypertension and diabetes and a higher mortality [51]. In their study, Grasselli et al. found that a history of chronic obstructive pulmonary disease, type 2 diabetes and hypercholesterolemia were independent risk factors associated with the mortality of patients admitted to an ICU due to COVID-19 in Lombardy [54]. Parohan et al. found an association between hypertension, cardiovascular disease, COPD, diabetes and cancer and a higher risk of death from COVID-19 [52]. Pugliese et al. and Tamura et al. both stressed that diabetes was a risk factor for COVID-19 progression to a critical illness as well as the development of acute respiratory distress syndrome, the need for mechanical ventilation or ICU admission and eventually death [56,57]. The findings from our study were similar to those presented above. We found that diabetes, a stroke history, renal failure, heart failure and COPD were associated with an increased mortality of COVID-19 patients.

In their study, Vila-Corcoles et al. found that pre-existing comorbid conditions (respiratory, renal or cardiac disease as well as hypertension and diabetes) and two symptoms—namely, dyspnoea and confusion—were significantly associated with an increased risk of ICU admission/mortality whereas symptoms such as muscle pain, headaches and anosmia were associated with a reduced risk of death [41]. Li et al. found that gastrointestinal symptoms such as vomiting, nausea and abdominal pain as well as respiratory symptoms such as chest pain and shortness of breath were associated with severe COVID-19 whereas end-stage organ failure and pneumonia were associated with mortality [46]. A study by Alizadehsani et al. demonstrated that symptoms of a COVID-19 infection such as a dry cough, ageusia, fever and anosmia were significantly associated with increased mortality [53]. Our study showed that shallow breathing and fever were associated with an increased mortality of COVID-19 patients. It is important to stress that the above-mentioned studies by Alizadehsani et al. and Zhang et al. both found that a fever was the most important symptom of COVID-19 [53,58]. In contrast, studies by, for example, DeBiasi et al. and Qin et al. did not find a significant association between a fever and a COVID-19 infection [59,60]. These findings demonstrated that there are no COVID-19-specific symptoms and may explain the results from our study concerning the symptoms of COVID-19 and mortality.

In the present study, we also analysed the relationship between the clinical parameters on admission and the prognosis of COVID-19 patients. Our analysis revealed that a higher heart rate and a higher respiratory rate were associated with an increased likelihood of death. A study by Mudatsir et al. demonstrated that high blood pressure and an increased respiratory rate were associated with severe COVID-19 [39]. Maloberti et al. found that the discharge heart rate was strongly related to the evidence of severe disease and the need for intensive care unit admission and/or mechanical ventilation [61].

The regression analysis carried out found that the haematocrit values and lymphocyte counts on admission were significant factors affecting the survival of COVID-19 patients. The analysis demonstrated that lower haematocrit values and lymphocyte counts were associated with an increased likelihood of death. A study by Mudatsir et al. demonstrated that increased leucocyte and neutrophil levels as well as decreased lymphocyte levels were associated with severe illness from COVID-19 [39] whereas Wang and Wang found that patients who died from COVID-19 had a significantly lower percentage of lymphocytes as well as significantly lower platelet counts and albumin levels compared with patients who survived [44]. In their study, Zhou et al. found that d-dimer levels greater than 1 μg/mL on admission were a risk factor for the mortality of adult COVID-19 patients. Moreover, the authors observed that increased levels of blood interleukin 6 (IL-6), cardiac troponin I and lactate as well as lymphopenia were more commonly seen in patients with a severe COVID-19 illness [18].

We would like to note that, according to the material analysed and our own experience, the key strategy aimed at reducing the load on traditional hospitals during the peaks of the pandemic was to set up temporary hospitals. The model most commonly adopted around the world involved converting fairgrounds or sports facilities into medical facilities. It is necessary to conduct further studies on the risk factors for severe illness and mortality from COVID-19, which are crucial in terms of the development of strategies for the treatment of patients, especially given the absence of an effective drug treatment.

The present report is the first Polish retrospective study of SARS-CoV-2 patients hospitalised at a temporary hospital at the National Stadium in Warsaw, which was set up as part of the strategy against COVID-19. We would like to stress that the present study has certain limitations. First, the data analysed came from only one, albeit the largest, of the temporary hospitals set up in Poland during the pandemic. Second, as the study was retrospective in nature, it was based on an analysis of hospital records, which had data gaps and were not completed in a uniform manner. Moreover, several laboratory test results were missing. We made every effort to ensure that these limitations did not undermine the quality of the study. Further studies on the treatment of COVID-19 patients are necessary to gain a better understanding of the subject in question and in particular to identify as many factors as possible affecting the prognosis of COVID-19 patients. Such studies would provide knowledge that could be used in practice in the treatment and care of COVID-19 patients. 

## 5. Conclusions

In conclusion, this is the first study concerning the hospitalisation of COVID-19 patients at a temporary hospital in Poland during the SARS-CoV-2 pandemic. Most patients were men and their median age was 64 years. The most commonly reported symptoms were a cough and fever and the most common comorbidities were hypertension and diabetes. The mortality rate of the patients was 8.9%. The analysis identified 14 factors that had a significant impact on the prognosis and mortality of the COVID-19 patients studied.

## Figures and Tables

**Table 1 ijerph-19-03932-t001:** Patient characteristics and analysis of the relationship between mortality and selected variables.

Variables	Total	Survivor	Non-Survivor	*p*-Value
	(*n* = 1749)	(*n* = 1593)	(*n* = 156)	
Sex, *n* (%)				
Female	715 (40.9)	661 (92.5)	54 (7.6)	0.095
Male	1034 (59.1)	932 (90.1)	102 (9.9)	
Age (years), Me (IQR)	64.0 (50.0–73.0)	63.0 (49.0–71.5)	72.4 (68.0–80.0)	0
BMI, Me (IQR)	29.0 (25.9–31.7)	29.0 (25.9–32.0)	27.6 (25.7–30.9)	0.079
Place of residence, *n* (%)				
Urban area	1498 (85.7)	1357 (90.6)	141 (9.4)	0.077
Rural area	251 (14.3)	236 (94.0)	15 (6.0)	
Symptoms, *n* (%)				
Cough	1014 (58.0)	928 (58.3)	86 (55.1)	0.45
Fever	891 (50.9)	824 (51.7)	67 (43.0)	0.036
Dyspnoea	879 (50.3)	791 (49.7)	88 (56.4)	0.107
Shallow breathing	494 (28.2)	443 (27.8)	51 (32.7)	0.196
Muscle pain	408 (23.3)	384 (24.1)	24 (15.4)	0.014
Headache	388 (22.2)	361 (22.7)	27 (17.3)	0.125
Loss/change in sense of taste	221 (12.6)	206 (12.9)	15 (9.6)	0.234
Diarrhoea	220 (12.6)	200 (12.6)	20 (12.8)	0.924
Loss/change in sense of smell	218 (12.5)	206 (12.9)	12 (7.7)	0.059
Chest pain	215 (12.3)	195 (12.2)	20 (12.8)	0.833
Sore throat	167 (9.6)	155 (9.7)	12 (7.7)	0.409
Rhinitis	115 (6.6)	108 (6.8)	7 (4.5)	0.27
Skin lesions	73 (4.2)	70 (4.4)	3 (1.9)	0.141
Chronic conditions, *n* (%)				
Hypertension	689 (39.4)	616 (38.7)	73 (46.8)	0.044
Diabetes	264 (15.1)	227 (14.3)	37 (23.7)	0.002
Myocardial infarction	139 (8.0)	121 (7.6)	18 (11.5)	0.082
Heart failure	123 (7.0)	100 (6.3)	23 (14.7)	0
Nicotine dependence	129 (7.4)	115 (7.2)	14 (9.0)	0.423
COPD	89 (5.1)	72 (4.5)	17 (10.9)	0.001
Cancer	66 (3.8)	55 (3.5)	11 (7.1)	0.024
Renal failure	61 (3.5)	46 (2.9)	15 (9.6)	0
Stroke history	59 (3.4)	47 (3.0)	12 (7.7)	0.002
Time of admission, *n* (%)				
7:00 a.m.–6:59 p.m.	967 (55.3)	883 (55.4)	84 (53.9)	0.704
7:00 p.m.–6:59 a.m.	782 (44.7)	710 (44.6)	72 (46.1)	
ICU, *n* (%)				
Yes	53 (3.0)	10 (0.6)	43 (27.6)	0
No	1696 (97.0)	1583 (99.4)	113 (72.4)	

Me: median; IQR: interquartile range; BMI: body mass index; COPD: chronic obstructive pulmonary disease; ICU: intensive care unit.

**Table 2 ijerph-19-03932-t002:** Clinical characteristics of patients and associations between mortality and selected clinical factors.

Variables	Total (*n* = 1749)	Survivor (*n* = 1593)	Non-Survivor (*n* = 156)	*p*-Value
Duration of hospitalisation, Me (IQR)	10.0 (7.0–15.0)	10.0 (7.0–14.0)	14.5 (9.0–22.0)	0.000
Extent of lung parenchyma affected (%), Me (IQR)	20.0 (9.0–35.0)	20.0 (8.0–35.0)	30.0 (10.0–50.0)	0.001
MEWS (points), Me (IQR)	3.0 (2.0–5.0)	3.0 (2.0–5.0)	5.0 (3.0–7.0)	0.000
Body temperature (°C), Me (IQR)	36.8 (36.4–38.5)	36.8 (36.4–38.5)	37.4 (36.4–39.0)	0.664
Heart rate (beats per minute), Me (IQR)	80.0 (70.0–90.0)	80.0 (69.0–90.0)	81.0 (77.0–93.0)	0.154
Systolic blood pressure (mmHg), Me (IQR)	130.0 (117.0–140.0)	130.0 (117.0–140.0)	129.0 (116.0–136.0)	0.607
Diastolic blood pressure (mmHg), Me (IQR)	78.0 (70.0–90.0)	79.0 (70.0–86.0)	72.0 (69.0–88.0)	0.476
MAP (mmHg), Me (IQR)	96.3 (87.0–104.8)	96.3 (87.0–104.7)	89.0 (85.0–109.0)	0.617
Respiratory rate (per minute), Me (IQR)	18.0 (17.0–21.0)	18.0 (17.0–20.0)	21.0 (18.0–25.0)	0.014
Saturation (%), Me (IQR)	96.0 (94.0–98.0)	96.0 (94.0–98.0)	93.0 (91.0–96.0)	0.003
Medical procedures, *n* (%)				
Pulse oximetry	1393 (79.7)	1272 (79.85)	121 (77.56)	0.499
Temperature measurement	1353 (77.4)	1236 (77.59)	117 (75.00)	0.461
Chest CT scan	785 (44.9)	717 (45.01)	68 (43.59)	0.734
Optiflow	583 (33.3)	501 (31.45)	82 (52.56)	0.000
Acid-base balance	489 (28.0)	445 (27.93)	44 (28.21)	0.943
Chest X-ray	469 (26.8)	426 (26.74)	43 (27.56)	0.825
Blood typing	437 (25.0)	392 (24.61)	45 (28.85)	0.243
Intubation	224 (12.8)	180 (11.30)	44 (28.21)	0.000
Arterial blood gas test	191 (10.9)	177 (11.11)	14 (8.97)	0.414
Mechanical ventilation	166 (9.5)	125 (7.85)	41 (26.28)	0.000
Antibiotic treatment	131 (7.5)	119 (7.47)	12 (7.69)	0.920
ECMO	5 (0.3)	0 (0.00)	5 (100.00)	1.000
Laboratory blood tests				
WBC count (thousand/µL), Me (IQR)	6.2 (4.6–8.3)	6.1 (4.6–8.2)	7.9 (6.5–12.0)	0.000
RBC count (million/µL), Me (IQR)	4.4 (4.1–4.8)	4.4 (4.1–4.8)	4.2 (3.9–4.6)	0.032
Haemoglobin (g/dl), Me (IQR)	13.4 (12.3–14.3)	13.4 (12.4–14.3)	13.1 (12.0–14.2)	0.502
Haematocrit (%), Me (IQR)	39.4 (36.7–42.1)	39.4 (36.6–42.1)	38.5 (36.7–41.8)	0.511
Platelet count (thousand/µL), Me (IQR)	231.0 (179.0–296.0)	231.0 (179.0–296.0)	214.0 (159.0–284.0)	0.486
Neutrophil count (thousand/µL), Me (IQR)	4.2 (2.8–6.)	4.2 (2.8–6.1)	7.0 (5.5–10.5)	0.000
Lymphocyte count (thousand/µL), Me (IQR)	1.1 (0.8–1.6)	1.1 (0.8–1.6)	0.6 (0.5–1.0)	0.000
Monocyte percentage (%), Me (IQR)	7.6 (5.4–10.23)	7.7 (5.5–10.3)	4.9 (3.1–6.5)	0.000
Eosinophil percentage (%), Me (IQR)	0.1 (0.1–0.5)	0.1 (0.1–0.5)	0.1 (0.1–0.1)	0.005

Me: median; IQR: interquartile range; MEWS: modified early warning score; MAP: mean arterial pressure; ECMO: extra corporeal membrane oxygenation; WBC: white blood cell; RBC: red blood cell.

**Table 3 ijerph-19-03932-t003:** Multivariate logistic regression analysis assessing the association between selected predictors and mortality of patients hospitalised with COVID-19.

Selected Predictors	Adjusted R^2^ = 0.245 *p* < 0.001				Exp(B) (Odds Ratio)	95% CI	
	B	SE	Wald	*p*			
ICU admission	4.098	0.364	26.537	0.000	60.238	29.495	123.024
MEWS	0.405	0.059	46.567	0.000	1.499	1.335	1.684
Mechanical ventilation	0.888	0.407	4.758	0.029	2.431	1.094	5.402
Diabetes	0.451	0.226	3.990	0.046	1.570	1.009	2.444
Renal failure	0.761	0.362	4.414	0.036	2.141	1.052	4.356
COPD	0.715	0.301	5.632	0.018	2.044	1.133	3.690
Stroke history	0.879	0.352	6.233	0.013	2.408	1.208	4.801
Heart failure	0.658	0.262	6.288	0.012	1.930	1.154	3.227
Fever	0.353	0.169	4.346	0.037	1.703	1.504	1.979
Shallow breathing	0.576	0.248	5.382	0.020	1.779	1.094	2.895
Heart rate	0.057	0.018	10.156	0.001	1.058	1.022	1.096
Respiratory rate	0.121	0.058	4.432	0.035	1.129	1.008	1.263
Haematocrit	−0.299	0.083	12.983	0.000	0.742	0.630	0.873
Lymphocyte count	−0.510	0.093	29.976	0.000	0.601	0.501	0.721

COPD: chronic obstructive pulmonary disease; ICU: intensive care unit; MEWS: modified early warning score.

## Data Availability

The data presented in this study are available on request from the corresponding author.

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
