# Peer review of "Analysis of Risk Factors for In-Hospital Death Due to COVID-19 in Patients Hospitalised at the Temporary Hospital Located at the National Stadium in Warsaw: A Retrospective Analysis"

_ijerph, 2022, doi:10.3390/ijerph19073932_

Round 1

Reviewer 1 Report

A very substantive, correctly written paper that meets the elements of a research paper.

I fully support its acceptance for publication, after very minor corrections. Congratulations on the idea for the paper 

- please add the place of your correspondence work
- I propose to change the topic to Analysis of risk factors for in-hospital death due to COVID-19 in patients hospitalized at the temporary hospital located at the National Stadium in Warsaw: a retrospective analysis 

Author Response

Dear Reviewer,

We wish to thank you very much for the time you have taken to read and review our paper.  We hope the changes we have made improved the overall quality of the paper in line with your expectations.

A very substantive, correctly written paper that meets the elements of a research paper.

I fully support its acceptance for publication, after very minor corrections. Congratulations on the idea for the paper

- please add the place of your correspondence work

We have changed this as suggested by the Reviewer

- I propose to change the topic to Analysis of risk factors for in-hospital death due to COVID-19 in patients hospitalized at the temporary hospital located at the National Stadium in Warsaw: a retrospective analysis

We have changed this as suggested by the Reviewer

The paper has been proofread again by a native speaker of English in a professional translation agency.

All the remarks and suggestions addressed in the text have been marked in red.

Reviewer 2 Report

Obviously, the idea of the manuscript is good. Nevertheless I have a few comments to authors:

1. Recently, some studies also indicated that diabetes mellitus can be risk factor of COVID-19 patients mortality, you could modify the discussion part.

2. The aim should be rephrased and more focused. For instance:  the aim of this study was to identify factors affecting mortality in COVID-19 patients at 
temporary hospital in Warsaw, Poland.

3. "Ethics" section should go after "statistical processing section".

4. In "the study design section" should be added in sentence "The documentation was analysed in order to obtain the following information: age and sex of the patients, date of admission and discharge, patients’ clinical parameters, medical procedures performed, laboratory test results, clinical symptoms and comorbidities. " information about outcomes.

5. A preferred option for analysis of risk factors on survival is Kaplan-Meier or Cox-regression analysis.

6. By what principle were the data for Table 2 selected if the median indicators of non-survivors do not clinically significantly differ? Statistical significance is achieved here due to the large number of observations. But the portrait of a non-surviving patient does not look clinically severe. It is possible that data selection was carried out at the point of hospitalization of the patient. In this case, this greatly shifts the focus, because you need to take the most revealing data during hospitalization.

7. The 95%CI for RBC count is too broad. So it can not be considered as a predictor.

8. Did you mean a stroke history in the table 5? Because now it looks like a presence of stroke during Covid-19.

9. The information "We would like to note that, according to the material analysed and our own experience, the key strategy aimed at reducing the load on traditional hospitals during the peaks of the pandemic is to set up temporary hospitals. The model most commonly adopted around the world involves converting fairgrounds or sports facilities into medical facilities. At the same time, it is necessary to conduct further studies on risk factors for severe illness and mortality from COVID-19, which are crucial in terms of the development of strategies for the treatment of patients, especially given the absence of effective drug treatment." should be transferred to discussion.

Author Response

Dear Reviewer,

We wish to thank you very much for the time you have taken to read and review our paper.  We hope the changes we have made improved the overall quality of the paper in line with your expectations.

Obviously, the idea of the manuscript is good. Nevertheless I have a few comments to authors:

  1. Recently, some studies also indicated that diabetes mellitus can be risk factor of COVID-19 patients mortality, you could modify the discussion part.

We have extended the Discussion section in accordance with the Reviewer’s suggestions.

  1. The aim should be rephrased and more focused. For instance:  the aim of this study was to identify factors affecting mortality in COVID-19 patients at temporary hospital in Warsaw, Poland.

We have changed this as suggested by the Reviewer

  1. "Ethics" section should go after "statistical processing section".

We have changed this as suggested by the Reviewer

  1. In "the study design section" should be added in sentence "The documentation was analysed in order to obtain the following information: age and sex of the patients, date of admission and discharge, patients’ clinical parameters, medical procedures performed, laboratory test results, clinical symptoms and comorbidities. " information about outcomes.

We have changed this as suggested by the Reviewer

  1. A preferred option for analysis of risk factors on survival is Kaplan-Meier or Cox-regression analysis.

We would like to thank the Reviewer for this suggestion. In this study we wanted to focus only on the identification of risk factors for in-hospital death due to COVID-19. An analysis of survival in COVID-19 patients using the Kaplan-Meier estimator will be the subject of our next publication, which is currently being prepared.

  1. By what principle were the data for Table 2 selected if the median indicators of non-survivors do not clinically significantly differ? Statistical significance is achieved here due to the large number of observations. But the portrait of a non-surviving patient does not look clinically severe. It is possible that data selection was carried out at the point of hospitalization of the patient. In this case, this greatly shifts the focus, because you need to take the most revealing data during hospitalization.

Data for Table 2 was obtained from the patients’ electronic medical records. Out of all the available results, the most recent test results recorded in the electronic system were chosen for the analysis. We are aware of the limitation of our analyses, which is mainly due to the retrospective nature of the study. Information about limitations is provided in the Limitation section.

  1. The 95%CI for RBC count is too broad. So it can not be considered as a predictor.

Thank you for the suggestion, RBC was removed

  1. Did you mean a stroke history in the table 5? Because now it looks like a presence of stroke during Covid-19.

Thank you for the remark, yes it was about a stroke history. We have modified in the manuscript.

  1. The information "We would like to note that, according to the material analysed and our own experience, the key strategy aimed at reducing the load on traditional hospitals during the peaks of the pandemic is to set up temporary hospitals. The model most commonly adopted around the world involves converting fairgrounds or sports facilities into medical facilities. At the same time, it is necessary to conduct further studies on risk factors for severe illness and mortality from COVID-19, which are crucial in terms of the development of strategies for the treatment of patients, especially given the absence of effective drug treatment." should be transferred to discussion.

We have changed this as suggested by the Reviewer

All the remarks and suggestions addressed in the text have been marked in red.

Reviewer 3 Report

Proposed paper is interesting and well written. However, some revisions are needed before it can be accepted for publication:

  • If I understand correctly the logistic regression showed are unadjusted. No covariates are indicated in the methods and in the results section. These numerous unadjusted models (tables 3 to 6) should be removed and only one multivariate model with all the variables found to be significant as covariates should be presented. This way only the real significant one can be presented. ICU admission need to be included as a covariates in the model since it is a very important determinants of mortality. Discussion should be changed accordingly.
  • Are creatinine and GFR available? in fact, as also recently published (10.3390/jcm10184108.), it is a strong mortality determinants. If unavailable please add this as a limitation. If available please add it into the model.
  • Some recent published papers with similar findings should be cited: 10.1007/s10389-021-01675-y; 10.2991/jegh.k.200928.001 ) and (10.1159/000522100 ).

Author Response

Dear Reviewer,

We wish to thank you very much for the time you have taken to read and review our paper.  We hope the changes we have made improved the overall quality of the paper in line with your expectations.

Proposed paper is interesting and well written. However, some revisions are needed before it can be accepted for publication:

If I understand correctly the logistic regression showed are unadjusted. No covariates are indicated in the methods and in the results section. These numerous unadjusted models (tables 3 to 6) should be removed and only one multivariate model with all the variables found to be significant as covariates should be presented. This way only the real significant one can be presented. ICU admission need to be included as a covariates in the model since it is a very important determinants of mortality. Discussion should be changed accordingly.

Thank you for the suggestion. We have changed this as suggested by the Reviewer

Are creatinine and GFR available? in fact, as also recently published (10.3390/jcm10184108.), it is a strong mortality determinants. If unavailable please add this as a limitation. If available please add it into the model.

We appreciate the Reviewer’s suggestion. However, information about GFR and creatinine levels was not available from the medical records obtained for the analysis. As suggested by the Reviewer, relevant information was added to the Limitation section.

Some recent published papers with similar findings should be cited: 10.1007/s10389-021-01675-y; 10.2991/jegh.k.200928.001 ) and (10.1159/000522100 ).

Thank you for the suggestion. We have cited the suggested papers.

The paper has been proofread again by a native speaker of English in a professional translation agency.

All the remarks and suggestions addressed in the text have been marked in red.

Round 2

Reviewer 2 Report

All comment have been satisfied. Good luck!

Author Response

Dear Reviewer,

We would like to express our gratitude for your time and a positive review of our study. We very much appreciate your acknowledgment of our work. 

Reviewer 3 Report

Authors replies to all the query raised and paper significantly improves. However, before it can be accepted other two point need to be corrected:

  • Renal failure, Heart Failure and Shallow breating at the multivariate analysis have a p-value that had the unity (1) falling into the 95% CI (es. Heart failure 0.860-2.787). This means that they are not statistically significant while a p-values < 0.05 is indicated. Is this an error? correct.
  • The finding that a higher HR is associated with mortality is really interesting and is a really novel finding. Also in our group we found higher HR to be related to ICU admission (J Clin Med. 2021 Nov 28;10(23):5590.). Please better discuss this point in the relative section.

Author Response

Dear Reviewer,

We wish to thank you very much for re-review our paper.  We hope the changes we have made improved the overall quality of the paper in line with your expectations.

Authors replies to all the query raised and paper significantly improves. However, before it can be accepted other two point need to be corrected:

    Renal failure, Heart Failure and Shallow breating at the multivariate analysis have a p-value that had the unity (1) falling into the 95% CI (es. Heart failure 0.860-2.787). This means that they are not statistically significant while a p-values < 0.05 is indicated. Is this an error? correct.

Thank you for your remark, yes it was a mistake. We have corrected  the table and the text of our article.

    The finding that a higher HR is associated with mortality is really interesting and is a really novel finding. Also in our group we found higher HR to be related to ICU admission (J Clin Med. 2021 Nov 28;10(23):5590.). Please better discuss this point in the relative section.

We have extended the Discussion section in accordance with the Reviewer’s suggestions.

All the remarks and suggestions addressed in the text have been marked in red.